# Lyophilized Drug-Loaded Solid Lipid Nanoparticles Formulated with Beeswax and Theobroma Oil

**DOI:** 10.3390/molecules26040908

**Published:** 2021-02-09

**Authors:** Hilda Amekyeh, Nashiru Billa

**Affiliations:** 1Department of Pharmaceutics, School of Pharmacy, University of Health and Allied Sciences, PMB 31, Ho, Ghana; hamekyeh@uhas.edu.gh; 2College of Pharmacy, QU Health, Qatar University, Doha, Qatar; 3Biomedical and Pharmaceutical Research Unit, QU Health, Qatar University, Doha, Qatar

**Keywords:** solid lipid nanoparticle, beeswax, theobroma oil, drug release, lyophilization

## Abstract

Solid lipid nanoparticles (SLNs) have the potential to enhance the systemic availability of an active pharmaceutical ingredient (API) or reduce its toxicity through uptake of the SLNs from the gastrointestinal tract or controlled release of the API, respectively. In both aspects, the responses of the lipid matrix to external challenges is crucial. Here, we evaluate the effects of lyophilization on key responses of 1:1 beeswax–theobroma oil matrix SLNs using three model drugs: amphotericin B (AMB), paracetamol (PAR), and sulfasalazine (SSZ). Fresh SLNs were stable with sizes ranging between 206.5–236.9 nm. Lyophilization and storage for 24 months (4–8 °C) caused a 1.6- and 1.5-fold increase in size, respectively, in all three SLNs. Zeta potential was >60 mV in fresh, stored, and lyophilized SLNs, indicating good colloidal stability. Drug release was not significantly affected by lyophilization up to 8 h. Drug release percentages at end time were 11.8 ± 0.4, 65.9 ± 0.04, and 31.4 ± 1.95% from fresh AMB-SLNs, PAR-SLNs, and SSZ-SLNs, respectively, and 11.4 ± 0.4, 76.04 ± 0.21, and 31.6 ± 0.33% from lyophilized SLNs, respectively. Thus, rate of release is dependent on API solubility (AMB < SSZ < PAR). Drug release from each matrix followed the Higuchi model and was not affected by lyophilization. The above SLNs show potential for use in delivering hydrophilic and lipophilic drugs.

## 1. Introduction

Interest in lipid-based drug delivery systems has intensified in the past decades because of their versatility. They can be presented in various forms to achieve desired therapeutic outcomes. Solid lipid nanoparticles (SLNs) comprise solid lipids dispersed in an aqueous surfactant solution, whereby the lipids used are typically solid at ambient temperature, thus allowing the particles to remain solid during drug release [1,2,3].

SLNs have good physical stability because the solid lipids help to impede drug leakage and degradation. Controlled drug release is possible with these carriers through careful selection of appropriate lipid combinations. Similar to other nanoparticulate systems, SLNs have a high surface area to volume ratio. Additionally, they can attain a high drug loading capacity, which gives them the potential to enhance the performance of lipophilic drugs [2]. SLNs may possess an adhesive ability that can improve oral drug delivery through delayed transit in the gut. For instance, following oral administration, they can adhere to the gut wall and release the drugs incorporated in them, thereby potentially improving bioavailability. Additionally, the solid lipids used in SLNs have an ability to promote particle absorption properties, which can be greatly beneficial in improving the bioavailability of lipophilic drugs [4,5,6]. SLNs have served as an alternative to polymeric particulate systems and are suitable carriers for hydrophobic drugs, peptides, proteins, and antigens; however, high encapsulation efficiencies for hydrophilic drugs remain a constraint due to incompatibility with the hydrophobic matrix [7].

Excipients used to prepare SLNs typically include solid lipids, emulsifiers, coemulsifiers, and aqueous media. Crucially, the solid lipids must be of high purity, safe, and ideally inexpensive. Solid lipids used may include vegetable fats (e.g., theobroma oil, shea butter), triglycerides (e.g., tristearin), partial glycerides (e.g., glyceryl behenate, glycerol monostearate), waxes (e.g., beeswax, cetyl palmitate, carnauba wax), and fatty acids (e.g., stearic acid), among others [3]. The physicochemical properties of lipids in SLN formulations, such as their tendency to exhibit polymorphism, crystallinity, melting point, and miscibility/solubility in solvents, must be evaluated prior to use since these are likely to be modified over time [8]. For instance, in storage above the melting points of the lipids, the SLN matrix will be destroyed. Matrix integrity can be assured with superior formulation characteristics during use and storage by using composite lipid matrices in SLNs. Generally, better in vivo tolerability can be obtained with natural fats than with synthetic fats. Moreover, natural lipids are reported to improve topical drug delivery by enhancing skin penetration and hydration, which augments the good permeation and occlusion properties that lipid nanoparticles possess [9,10].

Theobroma oil is an edible fat extracted from the seeds of *Theobroma cacao*. It typically contains palmitic, stearic, oleic, and linoleic acids [11]. Theobroma oil melts at body temperature (37 °C) and is extensively used in cosmetics, food, and pharmaceuticals, especially suppositories. Beeswax is a liquid secreted by special wax glands in the abdomen of young worker bees that solidifies on contact with air. It contains a mixture of hydrocarbons, free fatty acids, monoesters, diesters, triesters, hydroxy monoesters, hydroxy polyesters, and fatty acid polyesters. The melting temperature of beeswax is within 63–67 °C [12].

We previously conducted differential scanning calorimetry analyses on amphotericin B (AMB)-SLNs, paracetamol (PAR)-SLNs, and sulfasalazine (SSZ)-SLNs prepared with beeswax and theobroma oil (1:1) and found that the lipids were in their solid states within all three SLN types. There were only slight decreases in the melting points of theobroma oil (from 35.1 °C for the pure lipid to 32.9–33.4 °C within the SLNs) and beeswax (from 63.3 °C for the pure lipid to 61.3–61.9 °C within the SLNs), which indicated no modifications in the lipids within the SLNs. The reduction in melting point was mainly due to the high surface area of the SLNs [13]. We have also shown that AMB-SLN and PAR-SLN suspensions are stable after incubation in simulated gastric fluid for 2 h, followed by incubation in simulated intestinal fluid for another 2 h, as the SLNs had mean sizes and surface charges that were optimal for intestinal absorption after this treatment [14].

In the present study, we investigated whether the 1:1 beeswax–theobroma oil mixture, which is a relatively cheap, readily available, and edible lipid combination, is a suitable contender as a stable solid lipid matrix in nanoparticles containing different drugs after lyophilization. The parameters considered included particle size, zeta potential (ZP), and in vitro drug release, which are key indicators of stability [3]. This is important because several studies on SLN formulations focus on one drug; however, it may be useful if drugs with different physicochemical properties can be efficiently formulated into SLNs (suspension and lyophilized) using a simple matrix such as beeswax–theobroma oil (1:1) via the same method. Moreover, this can further point to method reproducibility and matrix efficiency. AMB, PAR, and SSZ, which have varying solubility profiles, were used as the model drugs in the present study. The predicted solubilities of these drugs in water, based on the ALOGPS 2.1 program by the Virtual Computational Chemistry Laboratory, are 0.0819, 4.15, and 0.0464 mg/mL, respectively. SLN lyophilization is a useful strategy that ensures that the final product is stable for the foreseeable shelf-life. Paradoxically, lyophilization may induce irrevocable stresses on formulations and compromise their physicochemical attributes, which in turn may affect the ultimate performance of the formulations. Thus, the effects of lyophilization on particle size and in vitro drug release were investigated to ascertain the impact of lyophilization stress-induced changes on the SLNs. The chemical structures of the model drugs are shown in Figure 1.

## 2. Results and Discussion

### 2.1. Physical Appearance of the Formulations

The colors of pure AMB, PAR, and SSZ powders are yellow, white, and dark yellow, respectively. The colors of the final products, suspension or lyophilized, mirrored those of the incorporated drugs. The encapsulation efficiencies of all the SLNs were >60% (60.7 ± 0.26, 78.4 ± 0.16, 91.2 ± 3.04% for the PAR-SLNs, SSZ-SLNs, and AMB-SLNs, respectively). The differences in encapsulation efficiencies were attributable to the aqueous solubilities of the drugs, since the solid lipid matrix favors higher encapsulation of hydrophobic drugs [13]. Freshly prepared SLNs were uniform in appearance without any signs of physical instability such as creaming, flocculation, coalescence, or particle sedimentation, whilst the lyophilized formulations were fluffy powders.

Lyophilization can be used to improve the physical and chemical stabilities of nanoparticle formulations. Moreover, lyophilized SLNs can be easily formulated into other dosage forms such as capsules [15,16], tablets [17,18], and films [19]. The formulations obtained show that beeswax–theobroma oil (1:1) can be used to prepare SLNs containing hydrophilic and hydrophobic drugs. In order to assure SLN stability, cryoprotectants are typically added to SLN dispersions at concentrations of 10–15% (of the total formulation) prior to lyophilization to reduce particle aggregation and effect reconstitution [20]. Trehalose is an efficient cryoprotectant for SLN formulations [21,22,23]; it was therefore used at a concentration of 10% in the present study.

### 2.2. Nanoparticle Tracking Analysis (NTA)

The particle concentration feature of NTA (nanoparticle tracking analysis) allows for the characterization of nanosized particles per mL of formulation, which in turn can be used for formulation optimization. The mean particle sizes obtained from the NTA study were 236.9 ± 19.4, 227.2 ± 16.9, and 212.2 ± 50.0 nm for the freshly prepared AMB-SLNs, PAR-SLNs, and SSZ-SLNs, respectively. The narrow size range is indicative of the potential of the solid lipid matrix and hence the versatility of the preparation method in producing SLNs for drugs with contrasting physicochemical properties. The size distribution by intensity graphs for the three SLNs are shown in Figure 2. The particle concentrations ranged from 2.05 × 10^8^ to 4.30 × 10^8^ per mL with PAR-SLNs exhibiting the highest number of particles, as judged by the intensity size range 150–275 nm, whilst SZZ-SLNs presented the lowest number of particles by intensity size range (84 and 169 nm). All three SLN types presented a minimal number of particles with sizes >500 nm. Considering that particle concentration was highest in the PAR-SLNs, it is possible that large (>500 nm) undetected particles in the AMB-SLN and SSZ-SLN preparations are aggregates of smaller particles. Moreover, it is reported that the presence of a few large particles in a sample reduces the number of small particles detected in NTA [24].

### 2.3. Dynamic Light Scattering (DLS) Analysis

The size distribution profiles (Figure 3) for the fresh SLNs were mostly unimodal with negligible proportions of particles >1000 nm, which were not detectable on the NTA (Figure 2). Therefore, despite the advantage of analyzing different particle populations using NTA, subsequent analyses were performed using DLS (dynamic light scattering) in order to obtain data on large-sized particles within the samples. Notwithstanding, there are some parallels in the size distribution profiles from the two particle size analyses that are therefore representative of the respective SLNs.

The z-average diameters were 210.1 ± 1.40, 206.5 ± 1.71, and 224.8 ± 3.31 nm for freshly prepared AMB-SLNs, PAR-SLNs, and SSZ-SLNs, respectively (Figure 4). Although the PAR-SLN formulation had the smallest z-average diameter in the DLS analysis, it had the highest particle concentration in the NTA. This is in line with a previous report in which a strong negative correlation was presented between particle size determined by DLS and particle concentration assessed by NTA [25]. There was no statistically significant difference between the z-average diameters of the three freshly prepared SLNs. Furthermore, the polydispersity index (PDI) values were also below 0.3, indicating mostly narrow size distributions.

The size profiles and PDIs for the SLNs stored for 24 months (4–8 °C) among the three formulations were similar to those obtained in the freshly prepared SLNs (Figure 3a,b), which is indicative of physical stability for this duration. Conversely, for the lyophilized samples, the AMB-SLNs showed a somewhat unimodal distribution, with some nominal inflections, whereas the PAR-SLNs and SSZ-SLNs showed mostly polymodal profiles. The z-average diameters of AMB-SLNs, PAR-SLNs, and SSZ-SLNs after storage (262 ± 3.5, 307.7 ± 2.7, and 256.6 ± 2.9 nm, respectively) and lyophilization (265 ± 40.7, 333.6 ± 56.8, and 283.6 ± 63.5 nm, respectively) were in all cases higher than those of respective fresh formulations (*p* < 0.05 in each instance). A careful analysis of Figure 4 shows that aside from the PAR-SLNs with a size increase of more than 100 nm as a result of storage or lyophilization, the AMB-SLNs and SSZ-SLNs showed increases of less than 60 nm. This could be attributed to the higher nanoparticle concentration in the PAR-SLN system, which favors collision resulting in particle aggregation and irreversible fusion of the particles [26].

The particle size range for all three formulations following lyophilization was 265 ± 40.7–333.6 ± 56.8 nm. A previous study using in vitro cell models as well as ex vivo and in situ rat ileum models revealed that nanoparticles having a size of 344.4 nm underwent intestinal transport via uptake by enterocytes and M cells. Additionally, it is reported that orally administered nanoparticles with sizes <500 nm can reach the systemic circulation [27,28]. Therefore, although lyophilization resulted in increases in the sizes of the AMB-SLNs, PAR-SLNs, and SSZ-SLNs, the final sizes of the formulations may be appropriate for oral administration. Nanoparticle size must be well suited for a target tissue or selected drug delivery route. Therefore, we believe that even when lyophilized, the SLNs can improve the oral absorption of hydrophobic drugs encapsulated within them. Consequently, the sizes of the lyophilized SLNs in this study will not preclude them from the benefits of nanoparticles following oral administration.

Storage temperature can affect the stability of drug formulations, particularly SLNs. Drug-loaded SLNs are more stable when stored at sub-ambient to above freezing temperatures (4–25 °C). This is because storage at higher or extremely low temperatures can result in physical instabilities, which may promote particle aggregation with increase in particle size and PDI, and modification to lipid crystalline structure, which may promote drug expulsion [29,30]. Aggregation of nanoparticles during storage at relatively high temperatures can be attributed to increased particle kinetic energy, which favors particle collisions. Moreover, high temperatures can distort the structural integrity provided by the surfactant film, leading to particle aggregation [29].

Furthermore, it is important to be cognizant of the melting point(s) of lipid(s) during SLN storage to ensure that the matrix remains solid, which guarantees drug retention. Storage of the fresh and lyophilized SLNs at 4–8 °C maintained the stability of the particles because the melting points of theobroma oil and beeswax are greater than 30 °C.

All the mean particle sizes obtained in this study for the three different drugs ranged from 206.5 nm to 333.6 nm, regardless of being freshly made, stored, or lyophilized.

ZP magnitude is a predictor of colloidal stability [3]. For instance, a minimum value of −30 mV signifies moderate stability. However, values greater than −60 mV indicate very good stability via electrostatic repulsions [31], which is important, particularly during storage. Electrostatic repulsions cause nanoparticles to repel each other, thereby avoiding aggregation. The ZP values obtained show that the surfaces of the three SLN preparations were negatively charged, contributed to by the anionic sodium cholate. The ZP magnitudes for all freshly prepared, stored, and lyophilized formulations in the present study were >60 mV as shown in Figure 5. These results indicate that the excipients and SLN formulation method conferred similar surface charge characteristics to the three formulations regardless of the drug load. These findings undoubtedly point to the dispersions being very stable systems irrespective of storage, type of drug, or lyophilization. Additionally, it is expected that the lyophilized SLNs will impart physical and chemical stability to the drug load prior to reconstitution since drug or excipient hydrolysis is potentially minimized.

### 2.4. In Vitro Drug Release Studies

Drug release from the freshly prepared SLNs has been reported previously [13]; however, cumulative drug release profiles from fresh and lyophilized SLNs over 8 h are compared in Figure 6. The percentage drug release values at the end of the study were 11.8 ± 0.4, 65.9 ± 0.04, and 31.4 ± 1.95% from fresh AMB-SLNs, PAR-SLNs, and SSZ-SLNs, respectively [13], and 11.4 ± 0.4, 76.04 ± 0.21, and 31.6 ± 0.33% from lyophilized AMB-SLNs, PAR-SLNs, and SSZ-SLNs, respectively. Using a *t*-test, differences in amount of drug released at 8 h were not statistically significant (*p* > 0.05) for each pair of fresh and lyophilized SLNs. Thus, lyophilization did not affect drug release from the 1:1 beeswax–theobroma oil SLN composite matrix and drug release is likely to remain identical between the fresh and lyophilized formulations following administration.

The lower release rate of AMB and SSZ from the SLNs is consistent with the fact that AMB [32] and SSZ [33] are class IV drugs (low aqueous solubility and low membrane permeability) according to the biopharmaceutics classification system, whereas PAR is assigned to class I [34] due to its high aqueous solubility and high membrane permeability. The drug release data show that most of the PAR is likely to be released from the SLNs in aqueous/biological media, whereas there is likely to be significant retention of AMB and SSZ within the SLNs prior to absorption after administration. The results also show that the release of the drugs from the SLN matrices is largely controlled by the solubility of the drug. Therefore, the higher release rate of PAR (>60% at 8 h) from both fresh and lyophilized SLNs can possibly be reproduced for drugs with similar characteristics.

It is likely that the formulation approach assures a uniform dispersion of the drugs within the SLN matrices and thus drug release is mainly controlled by their rate of diffusion from the matrix (see below). The relatively slow AMB and SSZ release from the SLNs is indicative of a slow diffusion rate from the lipid matrices into the surrounding medium. However, this slow in vitro drug release may be useful if controlled drug release is desirable, particularly following oral administration.

Depending on the polymorphic configuration, theobroma oil melts within 34–38 °C, whereas beeswax melts within 63–67 °C. The release study was performed at 37 °C. Therefore, the drug release pattern observed is attributable to initial melting of the theobroma oil, which prompts an initial drug release, whereas drugs located within the beeswax-dominated domains are retained longer and released more slowly, mostly by diffusion. Thus, the relative solubility of the drugs within the theobroma-oil-only, theobroma oil and beeswax, and beeswax-only matrices dictates the overall drug release characteristics. Furthermore, beeswax contains several hydroxyl groups and free fatty acids in its structure. Consequently, when beeswax is dispersed in an aqueous medium, it degrades and allows the ingress of water into its matrix. This is important for optimal release of the drugs because diffusion and hydrolytic degradation/erosion of matrices are key factors that precede drug release from SLNs [35,36].

The kinetics of drug release from each SLN type was investigated in order to understand the mechanism of drug release from the lipid matrices. Basically, drug release from matrices is categorized as diffusion-controlled, swelling-controlled, or chemically-controlled [37].

The R^2^ values obtained from the model equations are shown in Table 1. Usually, the model that provides the highest R^2^ value is considered the most appropriate for describing the type of release mechanism. As shown in Table 1, the R^2^ values are highest for the Higuchi model for all the freshly formulated and lyophilized SLNs, which is indicative of typical diffusion-controlled drug release. The Higuchi model indicates that drug release from a matrix involves simultaneous entry of the surrounding medium into the matrix, drug dissolution, and diffusion of the drug from the system [38]. Clearly, lyophilization did not affect the release mechanism of the drugs from the SLNs.

For the Korsmeyer–Peppas model, the release exponent *n*, which is used to assess the type of drug transport mechanism, was determined. The *n* is interpreted as follows: *n* = 0.5, 0.45 < *n* = 0.89, *n* = 0.89, and *n* > 0.89 are indicative of Fickian diffusion, non-Fickian (anomalous) transport, case II transport, and super case II transport, respectively [39]. However, the *n* value is valid only when applied to the initial 60% of drug release; therefore, we considered release data only from the PAR-SLNs, since AMB-SLNs and SSZ-SLNs achieved less than 35% cumulative release. The *n* values for fresh and lyophilized PAR-SLNs were 0.4963 and 0.6392, respectively, which designate non-Fickian diffusion. The model indicates that in Fickian diffusion, drug release is diffusion-controlled, whereas case II transport signifies that drug release is controlled by matrix erosion/relaxation. However, anomalous diffusion behavior is intermediate between the Fickian and case II types [39], indicating that both drug diffusion and lipid erosion/relaxation are involved. This anomalous mechanism is supported by our earlier assertion that drug release from the SLN matrix is preceded by melting of the theobroma oil and the rate of diffusion of the drug from the various matrix domains. Thus, the solubility of a drug within the matrices is a contributing factor to its release profile. The matrix relaxation possibly relates to changes in the volume-to-surface ratio of the nanoparticles due to high ingress of the release medium into the particles because PAR is hydrophilic [40,41].

### 2.5. Fourier-Transform Infrared Spectroscopy (FTIR) Analysis

Figure 7 shows the FTIR (Fourier-transform infrared spectroscopy) spectra for the pure drugs and their respective SLNs. In Figure 7a, the characteristic FTIR bands for pure AMB are at 3399 cm^−1^ for polyene C-H stretching and O-H stretching; 1692, 1570, and 1010 cm^−1^ for C=O stretching and N-H_2_ in-plane bending, polyene C=C stretching, and C-H out-of-plane bending, respectively. The peak at 1040 cm^−1^ can also be assigned to N-H_2_ out-of-plane bending and C-O-C symmetric stretching vibration in the pyranose ring of the compound. The peak at 1177 cm^−1^ results from C-O-C asymmetric stretching of the β-glycosidic linkage in the drug. Finally, the peak at 851 cm^−1^ is because of C-H bending in the pyranose ring [42,43,44]. The spectrum for AMB-SLNs (Figure 7b) reveals the presence of these characteristic peaks. However, the carbonyl stretching band at 1692 cm^−1^ appears shifted to a lower frequency (1652 cm^−1^) possibly because of the excipients used in the formulation because the spectra for PAR-SLNs (Figure 7d) and SSZ-SLNs (Figure 7f) also show similar bands at 1653 and 1635 cm^−1^, respectively.

Figure 7c shows the spectrum for pure PAR. The N-H stretching vibration is at 3326 cm^−1^. The bands at 3162, 1655, and 1565 cm^−1^ represent O-H group stretching vibration, the C=O group, and N-H in-plane bending, respectively. Additionally, the peaks at 1610, 1506, and 1442 cm^−1^ are characteristic of the aromatic ring, whereas the band at 1327 cm^−1^ represents O-H bending vibration. There is also a peak at 1260 cm^−1^ for C-N stretching vibration [45,46]. These bands can be identified in the spectrum for PAR-SLNs (Figure 7d), except those at 3326 and 3162 cm^−1^. However, the single large band at 3401 cm^−1^ in the SLN spectrum, which is characteristic of O-H, could have overshadowed the two bands. This is possibly because the N-H group is involved in an intermolecular hydrogen bonding.

Figure 7e shows the FTIR spectrum for pure SSZ. The broad band at 3438 cm^−1^ is attributable to phenolic and carboxylic O-H groups. The peaks at 1280, 1427, and 1618 cm^−1^ represent C=O bond, and symmetric and asymmetric stretching vibrations of the carboxylate moiety, respectively. The N=N group stretching in SSZ is detected at 1587 cm^−1^, whereas the peaks at 1359 cm^−1^ and 1173 cm^−1^ represent asymmetric and symmetric O=S=O stretching vibrations, respectively. Additionally, the peak at 1394 cm^−1^ corresponds to in-plane bending of the O-H group [47]. Figure 7f shows the spectrum for the SSZ-SLN formulation. The band at 1618 cm^−1^ in the SSZ spectrum appears shifted to a higher frequency (1635 cm^−1^) in the SLN spectrum. Interestingly, this band is at a similar position in the spectra for the other two SLNs and could therefore be largely due to functional group interactions in the excipients as it is not directly attributable to any of the excipients.

The marked resemblance between the three SLN spectra (Figure 7b,d,f) is because the preparations were similarly formulated, with the difference between them being the drugs. It also shows that the matrices are identical. The main differences observed in the spectra are slight shifts in frequencies and differences in peak shape and intensity when similar peaks are compared. This could be due to interactions that may have occurred among the excipients or between drugs and excipients.

In general, the peaks of several prominent functional groups in AMB, PAR, and SSZ are found in the respective SLN spectra, which confirms drug entrapment in the lipid matrix. Therefore, the spectra are not indicative of significant chemical interactions between the drugs and excipients.

## 3. Materials and Methods

### 3.1. Materials

AMB (Nacalai Tesque, Inc., Kyoto, Japan), PAR (Sigma-Aldrich, St. Louis, MO, USA), SSZ (Tokyo Chemical Industry Co. Ltd., Tokyo, Japan), beeswax (Acros Organics, Morris Plains, NJ, USA), and theobroma oil (JB Cocoa Sdn. Bhd., Johor, Malaysia) were obtained from the respective companies. Lecithin soy and sodium cholate were purchased from MP Biomedicals (Illkirch, France). Chloroform, ethyl acetate, and methanol were purchased from Fisher Scientific (Loughborough, UK). All reagents and solvents used were of analytical and high-performance liquid chromatography (HPLC) grades, respectively.

### 3.2. Preparation of SLNs

Separate drug-loaded SLNs were prepared using the emulsification-solvent diffusion technique as previously described [13]. Briefly, 50 mg drug, 120 mg lecithin, and 200 mg each of theobroma oil and beeswax were added to a mixture of chloroform and methanol (20 mL each). The ingredients were mixed and the solvents were evaporated at 50 °C. The drug–lipid mixture obtained was melted in ethyl acetate (20 mL, 70 °C). The mixture was added to sodium cholate solution (2.5% *w*/*v*, 40 mL) at 70 °C and homogenized (10,000 rpm, 6 min) using a T 25 homogenizer (IKA, Staufen im Breisgau, Germany). Water (60 mL, 70 °C) was then added slowly to the mixture with continuous stirring for 20 min, after which the organic solvent was evaporated. All three formulations were similarly prepared.

Samples from each formulation were lyophilized using an Alpha 1-2 LDplus freeze-dryer (Martin Christ, Osterode am Harz, Germany) for subsequent analyses. Trehalose was used as a cryoprotectant at a concentration of 10% during the lyophilization process. Another set of the formulations was stored at 4–8 °C for 24 months prior to analyses. Samples used in the FTIR analyses were formulated without trehalose.

### 3.3. Particle Size Analysis

The average size and size distribution of the SLNs were assessed by NTA and DLS as previously described [13,14]. NTA was performed using a NanoSight LM10 instrument (NanoSight, Amesbury, United Kingdom) equipped with a 640 nm laser. The formulations were diluted with deionized water and injected into the sample chamber with sterile syringes. DLS analysis was carried out using a Zetasizer Nano ZS^®^ (Malvern, UK). The samples were diluted with water, after which z-average, ZP, and PDI were determined. Triplicate measurements were performed for each sample and the data obtained have been expressed as mean ± standard deviation.

### 3.4. FTIR Analysis

FTIR spectra for the analysis of interactions between the drugs and excipients within the formulations were obtained on a Spectrum RX I spectrophotometer (Perkin Elmer, Waltham, MA, USA) over 4000–400 cm^−1^. Each sample was separately mixed with potassium bromide to obtain a homogeneous mixture. Thereafter, a well-formed disc of each sample was obtained by compressing the mixture at 5 tons for 5 min using a pellet press (Model 4350; Carver Inc., Jeffersonville, IN, USA) prior to the analysis.

### 3.5. Drug Release Studies

Cumulative in vitro drug release studies were performed as previously described [13]. Freshly prepared and lyophilized SLNs containing approximately 0.5 mg of drug were added to phosphate-buffered saline (PBS, pH 7.4) as the release medium and incubated at 37 °C in a WiseCube WIS-20^®^ shaking incubator (PMI-Labortechnik GmbH, Grafstal, Switzerland). The shaker was operated at 100 rpm and 1 mL aliquots were withdrawn (and replenished with fresh PBS) at 0, 1, 2, 4, and 8 h for analysis. SLNs were precipitated using 0.1 M HCl and centrifuged at 14,000 rpm for 10 min, after which a 20 μL aliquot of the supernatant was analyzed by HPLC (PerkinElmer, Inc., Waltham, MA, USA). The cumulative drug release with time was then plotted for each drug.

The kinetics of drug release from each SLN was determined by fitting the release data obtained into first order, Higuchi, Hixson–Crowell, and Korsmeyer–Peppas equations, shown in Table 2. The coefficients of determination (R^2^) were calculated from the linear curves obtained by regression analysis of each plot. The best kinetic model for the release data was determined by comparing the R^2^ values. For the Korsmeyer–Peppas model, the release exponent n, which is used to assess the type of drug transport mechanism, was also determined.

### 3.6. Data Analysis

Data analysis was done using GraphPad Prism (version 5; GraphPad Software, Inc., La Jolla, CA, USA). Paired *t*-test was used to analyze differences in the data. Statistical significance was considered at *p* < 0.05. Data have been expressed as mean ± standard deviation from triplicate measurements.

## 4. Conclusions

Three SLNs containing different model drugs were successfully formulated and exhibited good physical stabilities during storage for 24 months at 4–8 °C. Lyophilization resulted in a significant increase in particle size; however, the average size of the formulations remained <350 nm. Additionally, ZP, which is an important indicator of SLN stability, was not affected by lyophilization or storage for 24 months at 4–8 °C. For specific drugs, lyophilization did not alter the mechanism and rate of drug release from the SLN matrices, which was typically diffusion-controlled. The SLN matrix presented characteristic drug peaks in FTIR spectra confirming that the drugs may be present in an amorphous state and crucially free from interactions with the excipients. We may conclude that beeswax–theobroma oil (1:1) is an effective SLN matrix for encapsulating drugs with varying physicochemical properties and retains its physical stability when appropriately stored for up to two years.

## Figures and Tables

**Figure 1 molecules-26-00908-f001:**
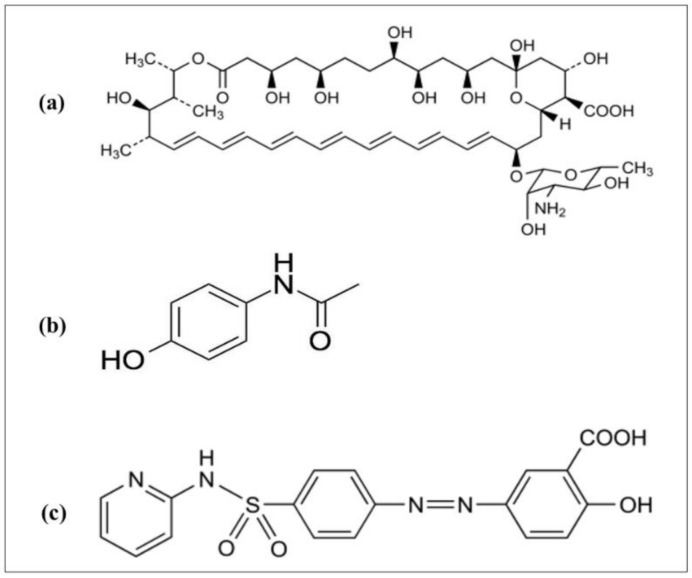
Chemical structures of (**a**) AMB (amphotericin B), (**b**) PAR (paracetamol), and (**c**) SSZ (sulfasalazine).

**Figure 2 molecules-26-00908-f002:**
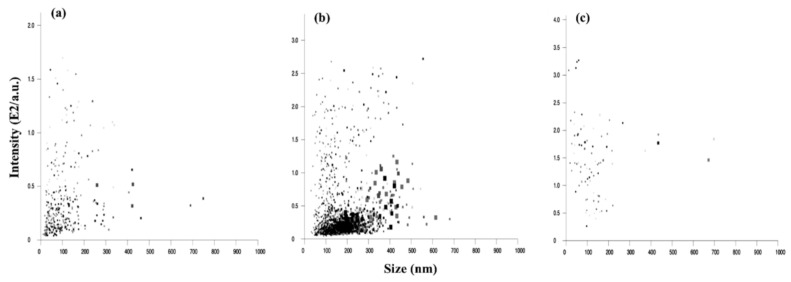
NTA (nanoparticle tracking analysis) size distribution by intensity graphs for fresh (**a**) AMB-SLNs (solid lipid nanoparticles), (**b**) PAR-SLNs, and (**c**) SSZ-SLNs (*n* = 3).

**Figure 3 molecules-26-00908-f003:**
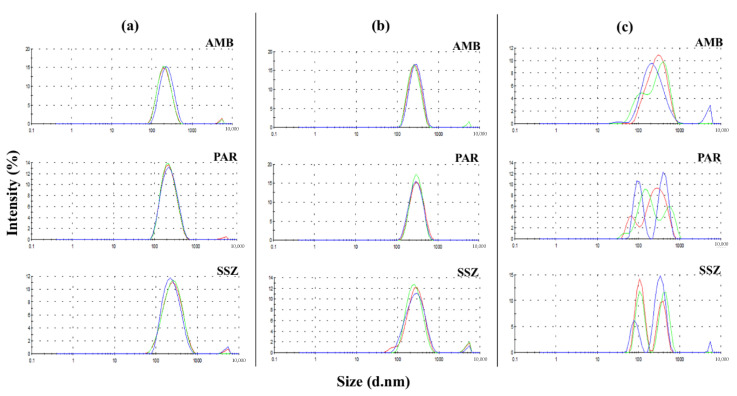
DLS (dynamic light scattering) size distribution by intensity graphs for (**a**) freshly prepared, (**b**) stored (24 months), and (**c**) lyophilized AMB-SLNs, PAR-SLNs, and SSZ-SLNs (*n* = 3).

**Figure 4 molecules-26-00908-f004:**
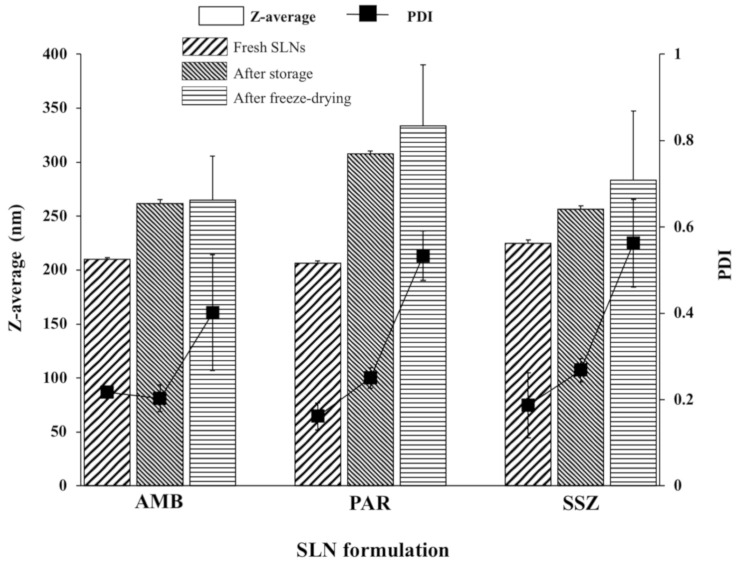
z-Average diameters and PDIs (polydispersity indexes) of freshly prepared, stored (for 24 months), and lyophilized AMB-SLNs, PAR-SLNs, and SSZ-SLNs (*n* = 3).

**Figure 5 molecules-26-00908-f005:**
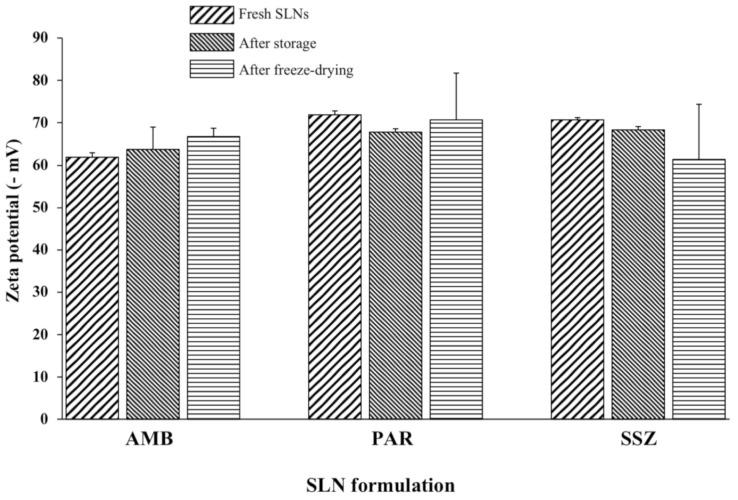
ZPs (zeta potentials) of freshly prepared, stored (24 months), and lyophilized AMB-SLNs, PAR-SLNs, and SSZ-SLNs (*n* = 3).

**Figure 6 molecules-26-00908-f006:**
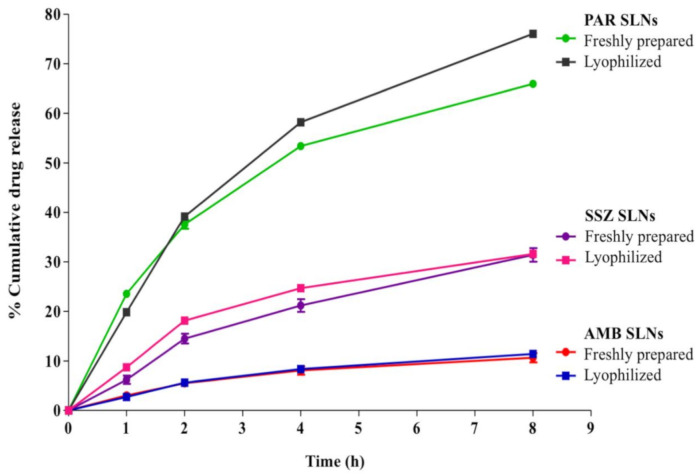
Cumulative drug release from freshly prepared and lyophilized AMB-SLNs, PAR-SLNs, and SSZ-SLNs in PBS (phosphate-buffered saline; pH 7.4).

**Figure 7 molecules-26-00908-f007:**
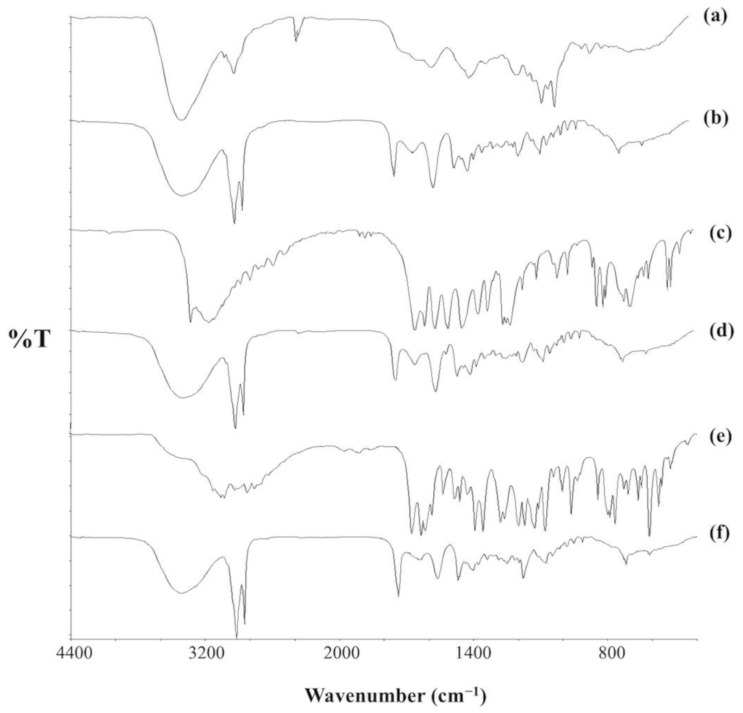
FTIR (Fourier-transform infrared spectroscopy) spectra for (**a**,**b**) AMB, (**c**,**d**) PAR, and (**e**,**f**) SSZ. (**a**,**c**,**e**) Pure drugs. (**b**,**d**,**f**) SLNs.

**Table 1 molecules-26-00908-t001:** Kinetics of in vitro drug release from AMB-SLNs, PAR-SLNs, and SSZ-SLNs.

Formulation	R^2^				*n* Value
	First Order	Higuchi	Hixson-Crowell	Korsmeyer–Peppas	
Fresh AMB-SLNs	0.8997	0.9889	0.8964	0.9704	0.5960 ^1^
Lyophilized AMB-SLNs	0.9206	0.9826	0.9174	0.9606	0.6740 ^1^
Fresh PAR-SLNs	0.9360	0.9858	0.9077	0.9728	0.4963
Lyophilized PAR-SLNs	0.9820	0.9825	0.9592	0.957	0.6392
Fresh SSZ-SLNs	0.9631	0.9743	0.9559	0.9563	0.7565 ^1^
Lyophilized SSZ-SLNs	0.8932	0.9765	0.8821	0.9297	0.6035 ^1^

^1^ data not applicable as drug release <60% during study period; AMB, amphotericin B; PAR, paracetamol; SLN, solid lipid nanoparticle; SSZ, sulfasalazine.

**Table 2 molecules-26-00908-t002:** Models for drug release kinetics.

Model	Expression
Zero order	Q = Kt
First order log	Q_R_ = Kt/2.303
Higuchi	$Q=K{\left(t\right)}^{\frac{1}{2}}$
Hixson–Crowell	Q013− Qt13=Kt
Korsmeyer–Peppas	log Q = log K + nlog t

Q, percentage cumulative release; Q_R_, percentage drug remaining; t, time in h; K, constant; n, release/diffusional exponent.

## Data Availability

The data presented in this study are available in article.

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
