# Peer review of "Lyophilized Drug-Loaded Solid Lipid Nanoparticles Formulated with Beeswax and Theobroma Oil"

_molecules, 2021, doi:10.3390/molecules26040908_

Round 1
Reviewer 1 Report
Since the aim of the paper is only to examine the stability of the SLN drug delivery system, it has to be characterized with appropriate methods. No direct experimental proof of the stability is presented; that is why my opinion is that this work is not appropriate for a highly ranked journal.
Author Response
Responses to Reviewers’ Comments
Reviewer 1
Since the aim of the paper is only to examine the stability of the SLN drug delivery system, it has to be characterized with appropriate methods. No direct experimental proof of the stability is presented; that is why my opinion is that this work is not appropriate for a highly ranked journal.
Response: We thank the reviewer for this comment. Our SLNs were prepared by adding the drugs to theobroma oil and beeswax (solid lipids) and dispersing the mixture in its liquid form in water using sodium cholate and lecithin as emulsifiers at high speed homogenization. This resulted in the formation of emulsion-based nanoparticles. We agree that there are several ways to assess nanoparticle stability. We have previously performed differential scanning calorimetry studies on the SLNs, as well as evaluated the SLNs after incubation in simulated gastrointestinal fluids. We have now clearly stated these in the manuscript and cited the references. Several studies have used changes in size and zeta potential as indicators of nanoparticle stability after lyophilization [Abdelwahed et al., 2006; Almalik et al., 2017; Phan and Haes, 2019], which we have done in our study. The in vitro release study was used to evaluate whether any lyophilization-induced stresses on the nanoparticles can affect drug release. Fourier-transform infrared spectroscopy is also valuable in assessing structural properties of lipids and drugs, which we have presented. Our major emphasis is that the three SLNs still have appreciable properties after lyophilization, in relation to the abovementioned parameters. This indicates that the solid lipid mixture (which is the same for all three preparations) may be a suitable matrix for encapsulating both hydrophilic and hydrophobic drugs in SLNs, whether as suspension or lyophilized.
References
Abdelwahed, W.; Degobert, G.; Stainmesse, S.; Fessi, H. Freeze-drying of nanoparticles: formulation, process and storage considerations. Adv. Drug Deliv. Rev. 2006, 58, 1688–1713. https://doi.org/10.1016/j.addr.2006.09.017.
Almalik, A.; Alradwan, I.; Kalam, M.A.; Alshamsan, A. Effect of cryoprotection on particle size stability and preservation of chitosan nanoparticles with and without hyaluronate or alginate coating. Saudi Pharm. J. 2017, 25, 861–867. https://doi.org/10.1016/j.jsps.2016.12.008.
Phan, H.T.; Haes, A.J. What does nanoparticle stability mean? J. Phys. Chem. C Nanomater. Interfaces 2019, 123, 16495–16507. https://doi.org/10.1021/acs.jpcc.9b00913.
Details of the revisions in the manuscript
Title, Page 1
We have modified the title of the manuscript from “Stable beeswax-theobroma oil matrices in solid lipid nanoparticles” to “Lyophilized drug-loaded solid lipid nanoparticles formulated with beeswax and theobroma oil” to ensure clarity.
Abstract
Minor spell check has been performed on the manuscript, as requested by the reviewer, and minor modifications have been made to improve the abstract for language and form. All changes are highlighted in yellow.
Page 1, Line 22; Page 7, Line 217
The abbreviation/symbol “P¥” has been removed from the abstract and main text since it is used only once in each instance.
Consistency in the use of mathematical symbols
We have ensured that there are no spaces around “±”, “<”, “>”, and “=”, except when used in equations or to directly indicate quantities/values, to ensure consistency (Lines 23, 103−104, 121, 129−130, 135, 145, 147, 158, 165−167, 197, 202, 212, 220, 236, 270−271, and 405).
Page 2, Line 60
Theobroma cacao has been italicized since it is a scientific name.
Page 2, Lines 67−86
We have now added information from our previous studies that show the stability of our formulations in differential scanning calorimetry studies and after incubation of the SLNs in simulated gastrointestinal fluids. These have been added in order to highlight that the present manuscript is in relation to SLN stability after lyophilization. The following paragraph has been added to the manuscript: “We previously conducted differential scanning calorimetry analyses on amphotericin B (AMB)-SLNs, paracetamol (PAR)-SLNs, and sulfasalazine (SSZ)-SLNs prepared with beeswax and theobroma oil (1:1) and found that the lipids were in their solid states within all three SLN types. There were only slight decreases in the melting points of theobroma oil (from 35.1°C for the pure lipid to 32.9–33.4°C within the SLNs) and beeswax (from 63.3°C for the pure lipid to 61.3–61.9°C within the SLNs), which indicated no modifications in the lipids within the SLNs. The reduction in melting point was mainly due to the high surface area of the SLNs [13]. We have also shown that AMB-SLN and PAR-SLN suspensions are stable after incubation in simulated gastric fluid for 2 h, followed by incubation in simulated intestinal fluid for another 2 h, as the SLNs had mean sizes and surface charges that were optimal for intestinal absorption after this treatment [14]” (Page 2, Lines 67−76).
We have also indicated that changes in particle size, zeta potential, and in vitro drug release after lyophilization can be used as indicators of SLN stability [3]. This has been indicated with some new statements, whereas some statements have also been modified to ensure clarity as follows: “In the present study, we investigated whether the 1:1 beeswax-theobroma oil mixture, which is a relatively cheap, readily available, and edible lipid combination, is a suitable contender as a stable solid lipid matrix in nanoparticles containing different drugs after lyophilization. The parameters considered included particle size, zeta potential (ZP), and in vitro drug release, which are key indicators of stability [3]. This is important because several studies on SLN formulations focus on one drug; however, it may be useful if drugs with different physicochemical properties can be efficiently formulated into SLNs (suspension and lyophilized) using a simple matrix such as beeswax-theobroma oil (1:1) via the same method. Moreover, this can further point to method reproducibility and matrix efficiency. AMB, PAR, and SSZ, which have varying solubility profiles, were used as the model drugs in the present study” (Page 2, Lines 77−86).
References
[3] Mehnert, W.; Mäder, K. Solid lipid nanoparticles: Production, characterization and applications. Adv. Drug Deliv. Rev. 2001, 47, 165–196. https://doi.org/10.1016/s0169-409x(01)00105-3.
[13] Amekyeh, H.; Billa, N.; Yuen, K.H.; Chin, S,L.S. A gastrointestinal transit study on amphotericin B-loaded solid lipid nanoparticles in rats. AAPS PharmSciTech 2015, 16, 871–877. https://doi.org/10.1208/s12249-014-0279-4.
[14] Amekyeh, H.; Billa, N.; Roberts, C. Correlating gastric emptying of amphotericin B and paracetamol solid lipid nanoparticles with changes in particle surface chemistry. Int. J. Pharm. 2017, 517, 42–49. https://doi.org/10.1016/j.ijpharm.2016.12.001.
Page 5, Lines 172−182
A new paragraph has been added to the manuscript, in which we have explained that the increase in particle size observed after lyophilization will not negatively affect the oral delivery of our formulations. We have cited two new references (which we have included in the reference list) to support this point. The paragraph added is as follows: “The particle size range for all three formulations following lyophilization was 265±40.7−333.6±56.8 nm. A previous study using in vitro cell models as well as ex vivo and in situ rat ileum models revealed that nanoparticles having a size of 344.4 nm underwent intestinal transport via uptake by enterocytes and M cells. Additionally, it is reported that orally administered nanoparticles with sizes <500 nm can reach the systemic circulation [27,28]. Therefore, although lyophilization resulted in increases in the sizes of the AMB-SLNs, PAR-SLNs, and SSZ-SLNs, the final sizes of the formulations may be appropriate for oral administration. Nanoparticle size must be well suited for a target tissue or selected drug delivery route. Therefore, we believe that even when lyophilized, the SLNs can improve the oral absorption of hydrophobic drugs encapsulated within them. Consequently, the sizes of the lyophilized SLNs in this study will not preclude them from the benefits of nanoparticles following oral administration“ (Page 5, Lines 172−182).
References
[27] He, C.; Yin, L.; Tang, C.; Yin, C. Size-dependent absorption mechanism of polymeric nanoparticles for oral delivery of protein drugs. Biomaterials 2012, 33, 8569–8578. https://doi.org/10.1016/j.biomaterials.2012.07.063.
[28] Woitiski, C.B.; Carvalho, R.A.; Ribeiro, A.J.; Neufeld, R.J.; Veiga, F. Strategies toward the improved oral delivery of insulin nanoparticles via gastrointestinal uptake and translocation. BioDrugs 2008, 22, 223–237. https://doi.org/10.2165/00063030-200822040-00002.
Due to the above addition, the number of references has increased to 46. The changes are highlighted in yellow.
Page 11, Line 361
“Zeta potential” has been abbreviated to ZP because the full meaning is now used for the first time earlier in the manuscript (Page 2, Line 80).
Page 12, Lines 409−410
The size limit of the nanoparticles after lyophilization has been indicated, and it has been emphasized that zeta potential, which is a key indicator of particle stability, was not affected by lyophilization. The statements now read as follows: “Lyophilization resulted in a significant increase in particle size; however, the average size of the formulations remained <350 nm. Additionally, ZP, which is an important indicator of SLN stability, was not affected by lyophilization or storage for 24 months at 4−8°C.”
Page 14, Lines 499−504
Two new references [27 and 28] (indicated above) have been added to the list of references.

Reviewer 2 Report
Dear authors you paper appears essential, but clear. I think you have enlarge the introduction in order to better focus your challenge. Furthermore the only in vitro drug results release seems to me lacking to support definitive conclusions about the perspective of your nanoparticles. Please justify at your best your choice and at least present clearly real perspectives about it. best regards
Author Response
Reviewer 2
Dear authors you paper appears essential, but clear. I think you have enlarge the introduction in order to better focus your challenge. Furthermore the only in vitro drug results release seems to me lacking to support definitive conclusions about the perspective of your nanoparticles. Please justify at your best your choice and at least present clearly real perspectives about it. best regards
Response: We thank the reviewer for this comment. We have now added statements on our previous findings to buttress our stability study of the nanoparticles with respect to lyophilization in the present study. These have been added to the introduction section of the revised manuscript (Page 2, Lines 67−86). The parameters assessed for all three SLN types were changes in particle size, zeta potential, and in vitro drug release. Several studies have assessed nanoparticle stability after lyophilization using changes in size and zeta potential [Abdelwahed et al., 2006; Almalik et al., 2017; Phan and Haes, 2019], which we have done in our study. The in vitro release was used to evaluate whether any lyophilization-induced stress on the nanoparticles can affect drug release. Our major emphasis is that the three SLNs still have appreciable properties after lyophilization, in relation to the abovementioned parameters as well as the Fourier-transform infrared spectra. Our results show that the solid lipid mixture (which is the same for all three preparations) may be a suitable matrix for encapsulating both hydrophilic and hydrophobic drugs in SLNs, whether as suspension or lyophilized.
References
Abdelwahed, W.; Degobert, G.; Stainmesse, S.; Fessi, H. Freeze-drying of nanoparticles: formulation, process and storage considerations. Adv. Drug Deliv. Rev. 2006, 58, 1688–1713. https://doi.org/10.1016/j.addr.2006.09.017.
Almalik, A.; Alradwan, I.; Kalam, M.A.; Alshamsan, A. Effect of cryoprotection on particle size stability and preservation of chitosan nanoparticles with and without hyaluronate or alginate coating. Saudi Pharm. J. 2017, 25, 861–867. https://doi.org/10.1016/j.jsps.2016.12.008.
Phan, H.T.; Haes, A.J. What does nanoparticle stability mean? J. Phys. Chem. C Nanomater. Interfaces 2019, 123, 16495–16507. https://doi.org/10.1021/acs.jpcc.9b00913.
Details of the revisions in the manuscript
Title, Page 1
We have modified the title of the manuscript from “Stable beeswax-theobroma oil matrices in solid lipid nanoparticles” to “Lyophilized drug-loaded solid lipid nanoparticles formulated with beeswax and theobroma oil” to ensure clarity.
Abstract
Minor spell check has been performed on the manuscript, as requested by the reviewer, and minor modifications have been made to improve the abstract for language and form. All changes are highlighted in yellow.
Page 1, Line 22; Page 7, Line 217
The abbreviation/symbol “P¥” has been removed from the abstract and main text since it is used only once in each instance.
Consistency in the use of mathematical symbols
We have ensured that there are no spaces around “±”, “<”, “>”, and “=”, except when used in equations or to directly indicate quantities/values, to ensure consistency (Lines 23, 103−104, 121, 129−130, 135, 145, 147, 158, 165−167, 197, 202, 212, 220, 236, 270−271, and 405).
Page 2, Line 60
Theobroma cacao has been italicized since it is a scientific name.
Page 2, Lines 67−86
We have now added information from our previous studies that show the stability of our formulations in differential scanning calorimetry studies and after incubation of the SLNs in simulated gastrointestinal fluids. These have been added in order to highlight that the present manuscript is in relation to SLN stability after lyophilization. The following paragraph has been added to the manuscript: “We previously conducted differential scanning calorimetry analyses on amphotericin B (AMB)-SLNs, paracetamol (PAR)-SLNs, and sulfasalazine (SSZ)-SLNs prepared with beeswax and theobroma oil (1:1) and found that the lipids were in their solid states within all three SLN types. There were only slight decreases in the melting points of theobroma oil (from 35.1°C for the pure lipid to 32.9–33.4°C within the SLNs) and beeswax (from 63.3°C for the pure lipid to 61.3–61.9°C within the SLNs), which indicated no modifications in the lipids within the SLNs. The reduction in melting point was mainly due to the high surface area of the SLNs [13]. We have also shown that AMB-SLN and PAR-SLN suspensions are stable after incubation in simulated gastric fluid for 2 h, followed by incubation in simulated intestinal fluid for another 2 h, as the SLNs had mean sizes and surface charges that were optimal for intestinal absorption after this treatment [14]” (Page 2, Lines 67−76).
We have also indicated that changes in particle size, zeta potential, and in vitro drug release after lyophilization can be used as indicators of SLN stability [3]. This has been indicated with some new statements, whereas some statements have also been modified to ensure clarity as follows: “In the present study, we investigated whether the 1:1 beeswax-theobroma oil mixture, which is a relatively cheap, readily available, and edible lipid combination, is a suitable contender as a stable solid lipid matrix in nanoparticles containing different drugs after lyophilization. The parameters considered included particle size, zeta potential (ZP), and in vitro drug release, which are key indicators of stability [3]. This is important because several studies on SLN formulations focus on one drug; however, it may be useful if drugs with different physicochemical properties can be efficiently formulated into SLNs (suspension and lyophilized) using a simple matrix such as beeswax-theobroma oil (1:1) via the same method. Moreover, this can further point to method reproducibility and matrix efficiency. AMB, PAR, and SSZ, which have varying solubility profiles, were used as the model drugs in the present study” (Page 2, Lines 77−86).
References
[3] Mehnert, W.; Mäder, K. Solid lipid nanoparticles: Production, characterization and applications. Adv. Drug Deliv. Rev. 2001, 47, 165–196. https://doi.org/10.1016/s0169-409x(01)00105-3.
[13] Amekyeh, H.; Billa, N.; Yuen, K.H.; Chin, S,L.S. A gastrointestinal transit study on amphotericin B-loaded solid lipid nanoparticles in rats. AAPS PharmSciTech 2015, 16, 871–877. https://doi.org/10.1208/s12249-014-0279-4.
[14] Amekyeh, H.; Billa, N.; Roberts, C. Correlating gastric emptying of amphotericin B and paracetamol solid lipid nanoparticles with changes in particle surface chemistry. Int. J. Pharm. 2017, 517, 42–49. https://doi.org/10.1016/j.ijpharm.2016.12.001.
Page 5, Lines 172−182
A new paragraph has been added to the manuscript, in which we have explained that the increase in particle size observed after lyophilization will not negatively affect the oral delivery of our formulations. We have cited two new references (which we have included in the reference list) to support this point. The paragraph added is as follows: “The particle size range for all three formulations following lyophilization was 265±40.7−333.6±56.8 nm. A previous study using in vitro cell models as well as ex vivo and in situ rat ileum models revealed that nanoparticles having a size of 344.4 nm underwent intestinal transport via uptake by enterocytes and M cells. Additionally, it is reported that orally administered nanoparticles with sizes <500 nm can reach the systemic circulation [27,28]. Therefore, although lyophilization resulted in increases in the sizes of the AMB-SLNs, PAR-SLNs, and SSZ-SLNs, the final sizes of the formulations may be appropriate for oral administration. Nanoparticle size must be well suited for a target tissue or selected drug delivery route. Therefore, we believe that even when lyophilized, the SLNs can improve the oral absorption of hydrophobic drugs encapsulated within them. Consequently, the sizes of the lyophilized SLNs in this study will not preclude them from the benefits of nanoparticles following oral administration“ (Page 5, Lines 172−182).
References
[27] He, C.; Yin, L.; Tang, C.; Yin, C. Size-dependent absorption mechanism of polymeric nanoparticles for oral delivery of protein drugs. Biomaterials 2012, 33, 8569–8578. https://doi.org/10.1016/j.biomaterials.2012.07.063.
[28] Woitiski, C.B.; Carvalho, R.A.; Ribeiro, A.J.; Neufeld, R.J.; Veiga, F. Strategies toward the improved oral delivery of insulin nanoparticles via gastrointestinal uptake and translocation. BioDrugs 2008, 22, 223–237. https://doi.org/10.2165/00063030-200822040-00002.
Due to the above addition, the number of references has increased to 46. The changes are highlighted in yellow.
Page 11, Line 361
“Zeta potential” has been abbreviated to ZP because the full meaning is now used for the first time earlier in the manuscript (Page 2, Line 80).
Page 12, Lines 409−410
The size limit of the nanoparticles after lyophilization has been indicated, and it has been emphasized that zeta potential, which is a key indicator of particle stability, was not affected by lyophilization. The statements now read as follows: “Lyophilization resulted in a significant increase in particle size; however, the average size of the formulations remained <350 nm. Additionally, ZP, which is an important indicator of SLN stability, was not affected by lyophilization or storage for 24 months at 4−8°C.”
Page 14, Lines 499−504
Two new references [27 and 28] (indicated above) have been added to the list of references.

Round 2
Reviewer 1 Report
Dear authors,
The reference (https://doi.org/10.1016/s0169-409x(01)00105-3.) given in your answer does not support all your claims in the manuscript. Please check section 5.3. and correct your claims and conclusions in the manuscript according to it.
Author Response
Response to Reviewer’s Comment
Reviewer 1
Dear authors,
The reference (https://doi.org/10.1016/s0169-409x(01)00105-3.) given in your answer does not support all your claims in the manuscript. Please check section 5.3. and correct your claims and conclusions in the manuscript according to it.
Response: We thank the reviewer for this pertinent comment. We have included an article (reference [31]) to support our discussions on zeta potential as a useful predictor of colloidal stability (section 2.3). We have also indicated the necessary values to clarify our statements and support our conclusions that our formulations have very good stability, whether freshly prepared or lyophilized. The revised statements now read, “ZP magnitude is a predictor of colloidal stability [3]. For instance, a minimum value of −30 mV signifies moderate stability. However, values greater than −60 mV indicate very good stability via electrostatic repulsions [31], which is important, particularly during storage” (page 6, lines 197−199). The references made to the article authored by Mehnert and Mäder (2001) in the manuscript are now clear.
[3] Mehnert, W.; Mäder, K. Solid lipid nanoparticles: Production, characterization and applications. Adv. Drug Deliv. Rev. 2001, 47, 165–196. https://doi.org/10.1016/s0169-409x(01)00105-3.
[31] Riddick, T.M. Control of Colloid Stability Through Zeta Potential; Livingston Publishing Company:
Wynnewood, PA, USA, 1968.
Details of the revisions in the manuscript
Page 6, lines 197−199
The statements have been revised to read “ZP magnitude is a predictor of colloidal stability [3]. For instance, a minimum value of −30 mV signifies moderate stability. However, values greater than −60 mV indicate very good stability via electrostatic repulsions [31], which is important, particularly during storage”.
Pages 6−9, 14 (lines 510−511)
A reference [31] has been added to the manuscript to ensure clarity with respect to our zeta potential data. Consequently, the number of references have increased from 46 to 47. Revisions have been made to the in-text citations (pages 6−9) and the article has been added to the reference list (page 14, lines 510−511).
Reviewer 2 Report
Dear I appreciate your revision. regards
Author Response
Thanks for the review.
This manuscript is a resubmission of an earlier submission. The following is a list of the peer review reports and author responses from that submission.
Round 1
Reviewer 1 Report
This manuscript is not novel. Studies on SLN are numerous and the technology has been around for a very long time. Please see as an example the paper: Eur J Pharm Biopharm . 2006 Nov;64(3):294-306 "Further characterization of theobroma oil-beeswax admixtures as lipid matrices for improved drug delivery systems". It is imperative that the authors clearly show how their results are enhancing our understanding of this system in drug delivery. Also experiments around the amorphous state and the characterization thereof and how it influence ultimate stability would make the manuscript more suitable for publication.Reviewer 2 Report
Dear Authors!
Unfortunately, I decided that this paper in the current form should be rejected; my main arguments are:
- The paper does not contain any examinations that are direct proof of formulation stability, like SEM/TEM images for morphological changes, XRD for structural changes, DSC for phase transition changes. Some of the examinations conducted and their results are indirect proof of stability, like size distribution and dissolution tests, and others are completely irrelevant, like IR studies.
- The manuscript is full of basic errors, like the classification of drug paracetamol or the claim that SLN has a solid lipid shell. Most of them are commented on and highlighted in the reviewed manuscript attached.
I am sure that paper can be reconsidered if direct proofs for stability are provided.

Reviewer 3 Report
The designations of the samples in the publication are not clear. Authors only use "AMB", "PAR", and "SSZ". I think it would be more correct to use abbreviations like "AMB-SLN" or similar to denote nanocarriers.
How can one explain such different degrees of encapsulation of particular drugs? It is necessary to explain.
What is the meaning of freeze-dried nanoparticles, since Their size and polydispersity change so much?